# Synergy between BRD9- and IKZF3-Targeting as a Therapeutic Strategy for Multiple Myeloma

**DOI:** 10.3390/cancers16071319

**Published:** 2024-03-28

**Authors:** Basudev Chowdhury, Swati Garg, Wei Ni, Martin Sattler, Dana Sanchez, Chengcheng Meng, Taisei Akatsu, Richard Stone, William Forrester, Edmund Harrington, Sara J. Buhrlage, James D. Griffin, Ellen Weisberg

**Affiliations:** 1Department of Medical Oncology, Dana-Farber Cancer Institute, 450 Brookline Avenue, Boston, MA 02215, USA; basudev_chowdhury@dfci.harvard.edu (B.C.); swati_garg@dfci.harvard.edu (S.G.); wei_ni@dfci.harvard.edu (W.N.); martin_sattler@dfci.harvard.edu (M.S.); dana.sanchez@stonybrook.edu (D.S.); taisei.akatsu@dfci.harvard.edu (T.A.); richard_stone@dfci.harvard.edu (R.S.); 2Department of Medicine, Harvard Medical School, Boston, MA 02115, USA; 3Novartis Pharma AG, 4056 Basel, Switzerland; william.forrester@novartis.com (W.F.); edmund.harrington@novartis.com (E.H.); 4Department of Cancer Biology, Dana-Farber Cancer Institute, Boston, MA 02215, USA; saraj_buhrlage@dfci.harvard.edu

**Keywords:** IMiD, BRD9, degrader, multiple myeloma, synergy, Ikaros

## Abstract

**Simple Summary:**

Multiple myeloma (MM) is an incurable disease affecting predominantly elderly people (>65 years of age). Despite improvement in the 10-year survival rate for MM patients due to the advent of medications, including immunomodulatory drugs (IMiDs) (such as pomalidomide and lenalidomide) and proteasome inhibitors, coupled with autologous stem cell transplant, more effective and safe therapies are still needed. Bromodomain-containing protein 9 (BRD9) has been shown to be important for the survival of MM cells. We discovered that drugs that block the function of BRD9 increase the effectiveness of IMiDs and can override resistance to these medications. Our findings suggest that the combined use of IMiDs and drugs that target BRD9 could potentially improve clinical outcomes for MM patients.

**Abstract:**

Progress in the treatment of multiple myeloma (MM) has resulted in improvement in the survival rate. However, there is still a need for more efficacious and tolerated therapies. We and others have shown that bromodomain-containing protein 9 (BRD9), a member of the non-canonical SWI/SNF chromatin remodeling complex, plays a role in MM cell survival, and targeting BRD9 selectively blocks MM cell proliferation and synergizes with IMiDs. We found that synergy in vitro is associated with the downregulation of MYC and Ikaros proteins, including IKZF3, and overexpression of IKZF3 or MYC could partially reverse synergy. RNA-seq analysis revealed synergy to be associated with the suppression of pathways associated with MYC and E2F target genes and pathways, including cell cycle, cell division, and DNA replication. Stimulated pathways included cell adhesion and immune and inflammatory response. Importantly, combining IMiD treatment and BRD9 targeting, which leads to the downregulation of MYC protein and upregulation of CRBN protein, was able to override IMiD resistance of cells exposed to iberdomide in long-term culture. Taken together, our results support the notion that combination therapy based on agents targeting BRD9 and IKZF3, two established dependencies in MM, represents a promising novel therapeutic strategy for MM and IMiD-resistant disease.

## 1. Introduction

Multiple myeloma (MM) represents approximately 20% of deaths from hematological malignancies. The National Cancer Institute found 34,470 newly diagnosed cases of MM, with 12,640 deaths estimated in the US in 2022, with a 57.9% 5-year survival rate. Over the past three decades, unfortunately, the overall death caused by the disease decreased only modestly [1]. 

Understanding the cell-signaling pathways in MM cells and the interaction of MM cells within the bone marrow (BM) microenvironment has led to the development of novel therapies [2]. New treatment strategies, such as proteasome inhibitors (PIs), immunomodulators (IMiDs), steroids, and targeted therapies, have changed the treatment options for multiple myeloma patients [3,4,5,6]. Cereblon (CRBN), a substrate receptor of the CRL4-CRBN E3 protein ligase complex, binds directly to IMIDs and then recruits and ubiquitinates Ikaros (IKZF1) and Aiolos (IKZF3), leading to their degradation at the 26S proteasome [7]. IKZF1 and IKZF3 downregulation leads to the downregulation of MYC [7,8,9], which is necessary for MM growth and viability [10]. MYC upregulation is linked to IMiD resistance [11], and CRBN aberrations, including acquired mutations of CRBN and associated genes, have been reported in a subset (22%) of IMiD-resistant MM patients [12,13]. CRBN loss and MYC upregulation are associated with a poor prognosis [11]. 

Most patients will be treated with maintenance therapy of IMiDs to extend the remission period. However, more than 90% of MM patients eventually relapse [14,15,16] while on IMiD therapy due to drug resistance [17]. The response rates of the relapsed–refractory population to new rounds of therapy decrease to the 25–30% range with approved regimens [18,19]. At present, pomalidomide, which has a more potent and less toxic profile than lenalidomide or thalidomide, is a standard-of-care treatment for relapsed/refractory MM patients [20,21,22]. There is an urgent need to develop novel therapeutic strategies to re-sensitize MM cells to IMiDs (especially pomalidomide) and improve the response rates in the relapsed–refractory population.

Bromodomain-containing protein 9 (BRD9) was recently found to be important for the maintenance of the transformed phenotype of various hematological malignancies. We and others observed a dependency on BRD9 in acute myeloid leukemia (AML), B-cell acute lymphoblastic leukemia (B-ALL), and MM [23,24,25,26,27]. 

Small molecule inhibitors of BRD9, which are based on a phenyl naphthyridone scaffold (BI7273 and BI9564), were shown to block the growth of AML cell lines [23,28]. We previously described a novel BRD9 bromodomain inhibitor, EA-89, and its more potent, degrader-based analog, QA-68, which incorporates the EA-89 warhead into a CRBN-targeting proteolysis-targeting chimera (PROTAC) [26]. We used QA-68, as well as BRD9 knockdown (KD) by RNA interference and CRISPR KO, as tools to demonstrate a robust dependency on BRD9 in the context of MM and acute leukemia [26]. We also demonstrated the potential for BRD9 targeting to potentiate the antitumor effects of chemotherapy drugs and targeted therapies as a novel treatment strategy for MM and acute leukemia [26]. Here, we show that the synergistic effects observed for the combination of BRD9 targeting and IMiDs against MM are in part mediated by Ikaros proteins, CRBN, and MYC and represent a novel potential approach for the treatment of MM. 

## 2. Materials and Methods

### 2.1. Chemical Compounds and Biologic Reagents

QA-68-ZU81, EA-89-YM35, and dBRD9-A (VD-83-AX41) were synthesized at Novartis Pharma AG, Basel, Switzerland, as previously described [26]. Lenalidomide was purchased from MedChemExpress (Junction, NJ, USA) (Cat. No.: HY-A0003). Pomalidomide was purchased from Millipore Sigma (Burlington, MA, USA) (P0018-5MG) and MedChemExpress (Cat. No.: HY-10984). 

### 2.2. Cell Lines and Cell Culture

H929 and RPMI-8226 MM cell lines were obtained from Dr. Kenneth Anderson (Dana-Farber Cancer Institute, Boston, MA, USA). Cell lines were cultured at 37 °C with 5% CO_2_ (2 × 10^5^–5 × 10^5^ cells/mL) in RPMI 1640 media, purchased from Gibco (Amarillo, TX, USA) (cat #11875-093). Media was supplemented with 10% fetal bovine serum (FBS), purchased from Gibco (Amarillo, TX, USA) (cat # 10437-028), and 1% penicillin/streptomycin (5000 U/mL), purchased from Gibco (Amarillo, TX, USA) (cat # 15070063). Culture media for H929 was supplemented with 2-mercaptoethanol (50 μM).

Cell lines were authenticated within six months of manuscript preparation via cell line short tandem repeat (STR) profiling (Molecular Diagnostics Core, Dana-Farber Cancer Institute). Tested cell lines matched >80% with lines listed in the DSMZ Cell Line Bank STR database (https://www.dsmz.de/catalogues/catalogue-human-and-animal-cell-lines/quality-assurance/identity-control/authentication-of-cell-lines.html, accessed on 1 July 2021). Cell lines were virus- and mycoplasma-free.

### 2.3. Cell Proliferation Studies

The Trypan Blue exclusion assay was performed for cell counting prior to seeding for CellTiter-Glo (Promega, Madison, WI, USA) proliferation assays. Studies were carried out according to the manufacturer’s instructions. Cell viability is graphed as the percentage of control (vehicle-treated) cells with error bars representing standard deviations (*n* = 4).

### 2.4. Drug Combination Studies

Single agents were added simultaneously at fixed ratios to cells. Cell number was expressed as a function of drug-treated, growth-inhibited cells versus DMSO vehicle control cells. Calcusyn software (Version 2.0) (Biosoft, Ferguson, MO, USA and Cambridge, UK), based on isobologram generation [29], was used to calculate antagonism or synergy as described previously [26].

### 2.5. Normal Peripheral Blood Mononucleated Cell (PBMC) Studies

Normal PBMCs were obtained through the approval of the Dana-Farber Cancer Institute Institutional Review Board from the Specimen Bank, Brigham and Women’s Hospital (Boston, MA, USA). Samples were Ficoll-purified to obtain mononuclear cells and cultured at 37 °C with 5% CO_2_ (2 × 10^5^–5 × 10^5^ cells/mL) in RPMI 1640 media, purchased from Gibco (Amarillo, TX, USA) (cat #11875-093). Media was supplemented with 10% fetal bovine serum (FBS), purchased from Gibco (Amarillo, TX, USA) (cat # 10437-028), and 1% penicillin/streptomycin (5000 U/mL), purchased from Gibco (Amarillo, TX, USA) (cat # 15070063). PBMCs were seeded at 20,000 cells/well for CellTiter-Glo (Promega, Madison, WI, USA) proliferation assays and treated with pomalidomide, EA-89, or a combination at the indicated concentrations for 4 days. Studies were carried out according to the manufacturer’s instructions. For flow cytometry, PBMCs were co-incubated with anti-human CD45 (cat# 304048) (Biolegend, San Diego, CA, USA) and FxCycle Violet (DAPI) (cat# F10347) (Thermo-Fisher Scientific, Waltham, MA, USA), for 15 min in the dark and the viable (DAPI-) CD45+ leukocytes were analyzed on BD LSRFortessa^TM^ X-20 Cell Analyzer at the DFCI Flow Cytometry Hematologic Neoplasia Core.

### 2.6. RNA-Seq Analysis

Total RNA was extracted from two independent biological replicates per each condition cell using the RNeasy Mini Kit (Qiagen, Venlo, The Netherlands) per the manufacturer’s instructions. Libraries were generated and sequenced on an Illumina NovaSeq 6000, targeting 40 million 100-bp read pairs per library at the Dana-Farber Cancer Institute Molecular Biology Core Facilities.

RNAseq analysis was performed using the VIPER snakemake pipeline [30], as described in Weisberg et al. [26]. Differential gene expression testing was performed by DESeq2 (v1.22.1) [31] and pathway analysis was conducted on genes that met the criteria of the Benjamini–Hochberg false discovery rate (FDR) < 0.05 and a log fold change value of > |1| by GeneCodis4 [32].

Gene set enrichment analysis (GSEA) was performed [33] on ranked gene lists that met the criteria of Benjamini–Hochberg false discovery rate (FDR) < 0.05 and a log fold change value of >|0.45| on the three sets (namely doxycycline only, pomalidamide only and the combination) of differentially expressed genes by WeGestalt [34] and significantly enriched (adjusted *p* value < 0.05) “hallmark” gene sets curated from The Molecular Signatures Database (MSigDB) [35] were determined.

### 2.7. Immunoblotting

Protein lysate preparation and immunoblotting were performed as previously described [36]. 

GAPDH antibody (14C10) (rabbit mAb, #2118), c-MYC antibody (D84C12) (rabbit mAb, #5605), Ikaros antibody (D6N9Y) (rabbit mAb, #14859), Aiolos antibody (D1C1E) (rabbit mAb, #15103), BRD9 antibody (E9R21) (rabbit mAb, #58906), and CRBN antibody (D8H3S) (rabbit mAb, #71810) were purchased from Cell Signaling Technology (Danvers, MA, USA). GAPDH antibody was used at a dilution of 1:2000 in 5% milk; all other antibodies were used at a dilution of 1:1000 in 5% milk.

### 2.8. Dox-Inducible BRD9 Knockdown (KD) and MYC Overexpression H929 Cells

Dox-inducible BRD9 H929 cells were developed as previously described [26]. MYC overexpressing H929 cells was generated similarly using MYC_pLX307, a gift from Dr. William Hahn and Dr. Sefi Rosenbluh (Addgene plasmid # 98363; http://n2t.net/addgene:98363, accessed on 27 March 2024; RRID:Addgene_98363).

For the BRD9 KD+ pomalidomide combination studies, we tested three H929 cell lines with different hairpins (BRD9 KD#1, BRD9 KD#2, and BRD9 KD#5) for the ability of doxycycline-induced BRD9 KD to potentiate pomalidomide. We carried out studies on different days (24 h, 48 h, 72 h, 96 h, and 6 days) with H929 cells as a control compared with each of the three different hairpins. Combination results for the 24 h time point were negative, and the results are not shown. Included in the main manuscript are proliferation/drug combination results for days 2, 3 and 4 and for the BRD9 KD#5 cell line as representative of all studies for which similar results were observed. These combination studies with BRD9 KD+ pomalidomide were performed to complement the combination studies performed with BRD9 degraders/inhibitors+IMiD and to validate that the effects observed with the drug combinations are BRD9-specific and on-target.

### 2.9. IKZF3-Overexpressing H929 Cells

#### H929 Stable Overexpression

For the stable overexpression of open reading frame (ORF) sequences, pLVX-EF1α-IRES-mCherry Vector was purchased from Takara (San Jose, CA, USA) (#631987), and the full-length sequence of EF1α promoter was replaced with custom synthesized EF1α core promoter. ORF containing cDNA sequence of IKZF1 (NM_006060.6 # OHu28071) and IKZF3 (NM_012481 # OHU21008D) in pCDNA vectors were purchased from GenScript and subcloned between EcoRI and SpeI sites into the custom pLVX-EF1α core-MCS-IRES-mCherry vector. To produce high titer lentivirus in Lenti-X 293T HEK cells from Takara (#632180), cells were transfected using Turbofect (Thermo#R0532), plasmid of interest and packaging plasmids PMD2.G (#12259) and psPAX2 (#12260) from Addgene. Virus supernatants were collected at 48 and 72 h post-transfection, filtered through 0.4micron filter, and ultra-centrifuged through 0.2micron filtered 20% sucrose before collection into 80microL of pure IMDM. Virus particles were titrated on MV4;11 AML cell line. H929 cells were spin-infected with empty vector control and IKZF3 overexpression virus at a multiplicity of infection (MOI) 5 at 1800 rpm for 30 min using 8 μg/mL polybrene. Cells were selected for mCherry production on day 3 by FACS. Then, protein overexpression by western blot and RT-PCR was expanded and validated before use in downstream assays.

For IKZF3 overexpression studies, we performed two independent studies, one with a BRD9 inhibitor (EA-89) and one with a BRD9 degrader (QA-68) plus and minus pomalidomide. Investigation of the rescue potential of IKZF3 overexpression was carried out to support IKZF3 mediation of synergy between BRD9-targeting drugs and IMiDs.

### 2.10. Development of Iberdomide-Resistant H929 Cells

H929 cells conferring resistance to the Cereblon E3 Ligase Modulator (CELMoD) iberdomide were developed following the culture of cells for 4 months in the presence of 10–20 nM iberdomide. In parallel, H929 cells were treated with an identical volume of vehicle (DMSO) and passaged in a similar fashion for the same length of time.

## 3. Results

### 3.1. BRD9 Inhibitor or Degrader Treatment Potentiates Effects of Immunomodulatory Drugs (IMiDs)

MM cells are dependent on IKZF3 and MYC for growth. Public CRISPR dependency data (depmap.org) identifies IKZF3 and MYC as dependencies in MM cell lines (*n* = 18), compared to cell lines derived from various other cancers (Appendix A). We hypothesized that co-targeting IKZF3 and BRD9 might enchance growth suppression of naïve and drug-resistant MM cells since each approach leads to the downregulation of MYC [7,8,9,26]. To test this notion, H929 or RPMI-8226 cells were treated with lenalidomide or pomalidomide alone and combined with either the BRD9 inhibitor, EA-89, or the BRD9 degrader, QA-68 (which is comprised of EA-89 linked to lenalidomide). Consistent with previously reported findings [26], EA-89 or QA-68 treatment potentiated the growth inhibitory effects of lenalidomide or pomalidomide in both lines (Figure 1A–D and Appendix A), with Calcusyn combination indices suggestive of synergy across a range of drug concentrations (Figure 1E). Importantly, the effects of pomalidomide and EA-89, alone and combined, led to substantially more cell killing of H929 cells than PBMCs derived from a healthy donor when both cell populations were treated in parallel (Figure 1F). These results suggest a therapeutic window for the combination of a BRD9 targeting agent and IMiD against MM cells. CD45+ leukocytes derived from PBMCs of a healthy donor were similarly treated with either agent alone or the combination, and no drug combination effect was observed (Figure 1G). 

At the protein level, the combination of EA-89 and pomalidomide led to a stronger reduction in MYC levels as compared to pomalidomide-only or EA-89-only-treated cells (Figure 2A, Appendix A). In comparison, we observed pomalidomide targets, Ikaros family proteins, such as IKZF1 and IKZF3, to be downregulated as well. however, only to a slightly greater extent, in response to the combination of pomalidomide and EA-89 compared to each agent alone (Figure 2A, Appendix A). CRBN levels, in contrast, were upregulated in combination-treated cells as compared to pomalidomide-only-treated or EA-89-only-treated cells (Figure 2A, Appendix A).

We compared the effects of lenalidomide and EA-89 as single agents on IKZF3 and MYC protein expression in H929 cells versus the effects of QA-68. The effects of lenalidomide (Figure 2B and Appendix A) and EA-89 (Figure 2C and Appendix A) on IKZF3 and MYC were weaker than those of QA-68 (Figure 2D and Appendix A). For comparison, pomalidomide downregulated IKZF3 to a similar extent to QA-68; however, pomalidomide downregulated MYC to a lesser extent than QA-68 (Figure 2D,E and Appendix A). Of note, EA-89 was not observed to influence CRBN protein levels (Figure 2C).

### 3.2. BRD9 Knockdown (KD) Potentiates Effects of Pomalidomide against Growth of Multiple Myeloma (MM) Cells

Doxycycline-inducible BRD9 KD in H929 cells led to the inhibition of cell growth, supporting the notion of a BRD9 dependency in MM (Figure 3A,B and Appendix A). Doxycycline-inducible BRD9 KD potentiated the effects of pomalidomide against H929 cells (Figure 3C,D and Appendix A). These results validate the contribution of BRD9-targeting to the synergy observed between BRD9 inhibitors or degraders combined with IMiDs and support the notion that BRD9 inhibitor/degrader effects are on-target.

### 3.3. IKZF3 or MYC Overexpression in H929 Cells Partially Reverses Synergy between BRD9-Targeted Agents and Pomalidomide

Overexpression of IKZF3 in H929 cells led to a partial reversal of the effects of pomalidomide alone and the effects of the combination of EA-89 plus pomalidomide or QA-68 plus pomalidomide (Appendix A). In addition, MYC overexpression in H929 cells partially reversed the effects of EA-89 alone, pomalidomide alone, and the combination of EA-89 combined with pomalidomide (Appendix A). These results, combined with IKZF3 protein downregulation in combination-treated cells (Figure 2A, Appendix A), suggest that IKZF3 plays at least a partial role in the synergistic interaction between BRD9 inhibitors or degraders and pomalidomide. In addition, these results validate the importance of downregulation of MYC in response to EA-89 or pomalidomide treatment and suggest that MYC may contribute to the molecular mechanism of synergy observed between BRD9 targeting agents and IMiDs. Furthermore, the decreased sensitivity of MYC-overexpressing MM cells to pomalidomide is consistent with the fact that the upregulation of MYC protein has been associated with IMiD resistance [11].

### 3.4. Inhibitory Effects of Pomalidomide Combined with Targeted Loss of BRD9 on Signaling Pathways in Multiple Myeloma

RNA-seq analysis was carried out on doxycycline-inducible H929 cells treated with doxycycline alone, pomalidomide alone, or a combination of doxycycline and pomalidomide for 24 h. GSEA enrichment analysis was performed and it revealed that E2F and MYC targets were among the top downregulated genesets associated with BRD9 KD plus pomalidomide (Figure 4A,B). MYC and E2F are critical transcription factors that can collaborate in the context of regulating genes involved in DNA replication and cell cycle progression [37,38,39,40]. We selected and validated enhanced downregulation of a panel of MYC target genes and/or E2F target genes in BRD9 KD plus pomalidomide-treated MM cells (Figure 4C and Appendix A). Not surprisingly, top downregulated pathways associated with BRD9 KD plus pomalidomide included cell cycle and cell division, as well as DNA replication, DNA repair, chromosome segregation, and kinetochore assembly (Figure 5A,B and Appendix A). Importantly, BRD9 KD plus pomalidomide combination-treated cells showed a substantially higher number of downregulated genes compared to either treatment alone (Appendix A). We validated the downregulation of EGR1 and INHBE, genes playing a role in cell growth and proliferation [41,42,43], in H929 following BRD9 knockdown along with pomalidomide (Figure 5C, left and middle panels).

The transcription factor, EGR2, associates with and regulates MYC [44,45] and, like EGR1, is involved in TNF-alpha signaling [46,47], which was identified as a top downregulated pathway in association with the combination of BRD9 KD plus pomalidomide (Figure 4A and Figure 5C, right panel). MYC induces EGR2 [44], and conversely, in effector T cells, EGR2 expression enhances clonal expansion and supports the regulation of expression of genes associated with proliferation, including MYC and MYB [45]. E2F transcription factors were identified as top suppressed pathways in response to BRD9 KD plus pomalidomide (Figure 5D,E). Genecodis Regulatory Transcription Factor Analyses (DoRothEA Regulons Evidence level: A, B, and C) predicted the transcription factor (TF) activities of MYC and E2F activities based on the gene expression of their targets (i.e., TF regulon). TF activities of MYC and E2F were reduced upon the combination treatment based on the gene expression of their targets (i.e., TF regulon).

### 3.5. Stimulatory Effects of Pomalidomide Combined with Targeted Loss of BRD9 on Signaling Pathways in Multiple Myeloma

Top stimulated pathways identified for BRD9 KD and pomalidomide-treated H929 cells include immune system process, cell adhesion, and inflammatory response (Figure 6A,B). Among genes upregulated to a greater extent in doxycycline- and pomalidomide-treated H929 BRD9 KD#1, H929 BRD9 KD#2, and H929 BRD9 KD#5 cells, compared to doxycycline or pomalidomide alone, is GIMAP4, which plays a role in the immune response [48] (Figure 6C). The STAT2 signaling pathway, associated with the immune response [49], was revealed to undergo elevated TF regulon activity in response to doxycycline-induced BRD9 KD coupled with pomalidomide (Figure 6D,E). 

### 3.6. BRD9 Targeting Overrides Immunomodulatory Drug (IMiD) Resistance

Iberdomide-resistant H929 cells were developed following long-term culture in the presence of 10–20 nM iberdomide (Figure 7A). Consistent with previous reports [11,50], we observed decreased expression of CRBN as an established mechanism of resistance to IMiD (Figure 7B, upper panel, and Appendix A). In these resistant cells, MYC expression levels are upregulated (Figure 7B, upper panel, and Appendix A), which is consistent with previous reports of MYC upregulation being associated with IMiD resistance [11]. In addition, IKZF3 protein levels were observed to be slightly elevated in iberdomide-resistant cells compared to parental cells (Figure 7B, upper panel, and Appendix A). These results support a previous report showing impairment of MYC and IKZF3 downregulation by lenalidomide in lenalidomide-resistant MM cells [50]. We did not observe EA-89 as a single agent to affect levels of CRBN protein in parental or iberdomide-resistant H929 cells (Figure 2C and Figure 7B, lower panel).

Iberdomide-resistant H929 cells were less sensitive to pomalidomide and another CELMoD, mezigdomide, compared to parental, DMSO-treated control cells (Figure 7C and Appendix A). The combination of EA-89, the BRD9 bromodomain inhibitor, and pomalidomide tested against both parental H929 (Figure 7D, left panel) and iberdomide-resistant H929 cells (Figure 7D, right panel) caused a leftward shift in the dose-response curve compared to EA-89 alone or pomalidomide alone. The drug combination against IMiD-resistant H929 cells (Figure 7D, right panel) was observed to be as potent as pomalidomide only against parental H929 cells (Figure 7C,D, left panel). These results suggest that treatment with the combination of a BRD9 inhibitor along with pomalidomide can override IMiD resistance. This combination increases CRBN and leads to the loss of MYC and IKZF3 (Figure 2A, Appendix A). This is significant as elevated MYC and IKZF3 protein levels and downregulation of CRBN protein levels are associated with IMiD resistance in our study and previous studies [11,50,51].

Taken together, these results support BRD9 targeting combined with IMiDs as a unique and novel approach to treating MM. This study also introduces the potentiation of IMiDs by BRD9 targeting as a potential strategy to override IMiD-resistant disease.

## 4. Discussion

MM is a malignancy characterized by a high percentage of clonal bone marrow plasma cells and accounts for around 10% of all hematologic cancers [52]. The overall 5-year survival rate for MM has been improved to 49–56% due to novel therapeutics, including proteasome inhibitors [53] and IMiDs, the latter, which kill transformed cells by targeting proteins for degradation through binding to CRBN, the substrate receptor of the CRL4-CRBN E3 ubiquitin ligase [54]. IMiDs effectively delay the onset of MM recurrence; however, challenges remain, including drug resistance and a high likelihood of relapse [14,15,16,17]. New generation IMiD analogs called CELMoDs, such as iberdomide and mezigdomide, are under clinical investigation to enhance the degradation of IKZF1/3 by increasing the protein-binding potential of CRBN [55]. However, while IMiDs have been highly successful medicines in MM, the development of resistance suggests additional strategies are needed to potentiate efficacy and raise the potential for higher remission rates and longer duration of response.

Using two established MM cell lines, H929 and RPMI-8226, we show that targeting IKZF1/3 and BRD9, two established dependencies in MM, leads to synergistic effects on viability. This synergy is associated with decreases in MYC protein. Consistent with this, treatment of MM cells with the lenalidomide-based BRD9 degrader, QA-68, led to greater downregulation of MYC than the corresponding BRD9 inhibitor, EA-89. This effect was also seen following the shRNA knockdown of BRD9. We observe that overexpression of IKZF3 or MYC in MM partially reversed the synergy between the agents.

The IMiD-based BRD9 degrader, dBRD9a [56,57], comprised of a BRD9 inhibitor coupled to an IMiD, was found to be selectively toxic toward primary MM cells. However, it was nontoxic to primary cells from a healthy donor at concentrations in the range of 100–1000 nM [26]. These selectively active concentrations of dBRD9a were comparable to, or higher than, combined concentrations shown in the present study to be synergistic for pomalidomide (75 nM)+ EA89 (BRD9 inhibitor) (750 nM) or pomalidomide (50 nM) + QA-68 (IMiD-based BRD9 degrader) (5 nM) against H929 MM cells, as shown in Figure 1B and Figure 1C, respectively. QA-68, comprised of EA-89 and an IMiD, was also shown [26] to be selectively toxic toward transformed (AML) cells at concentrations up to 1000 nM with a corresponding lack of toxicity toward normal bone marrow cells. Bone marrow colony assay results were similar and suggested selective toxicity of QA-68 toward transformed cells as compared to normal cells [26]. Also observed was a lack of toxicity of QA-68 against normal PBMCs at concentrations up to 10,000 nM [26]. Similarly, in the present study, we show that PBMCs from a healthy donor were insensitive to the same concentrations of EA-89 and pomalidomide, alone and combined, that were toxic toward H929 cells. These results, taken together, suggest that the combination of BRD9 targeting and IMiD treatment has a broad therapeutic window and is likely to be clinically translatable.

Transcriptional profiling revealed that BRD9 KD coupled with pomalidomide interferes with cell cycle and division, DNA replication and repair, chromosome segregation, and kinetochore assembly. The number of downregulated genes in BRD9 KD plus pomalidomide combination-treated cells was markedly higher compared to either treatment alone, suggesting greater perturbation of the transcriptional signature that may account, at least in part, for the overall higher potency of the combination. Enrichment analysis revealed MYC as one of the top downregulated pathways in association with BRD9 KD plus pomalidomide. Importantly, MYC and E2F, another top downregulated pathway identified by enrichment analysis, can promote cell cycle progression [37,38,39,40]. To substantiate findings from enrichment analysis, we validated increased downregulation of a panel of MYC target genes, including CDC45, MYC, LDHA, RFC4, and MAD2L1 [58,59,60,61], and E2F target genes, including CHEK1, DSCC1, DEPDC1, MMS22L, TOP2A, and MAD2L1 [62,63,64,65,66,67], in BRD9 KD plus pomalidomide-treated MM cells. 

To verify the results of transcriptional profiling and pathway analysis highlighting pathways reflecting growth suppression observed for BRD9 targeting plus IMiDs and loss of MYC, we validated the downregulation of a panel of selected genes. For example, INHBE, which plays a role in cell proliferation [41,42], was identified as being more downregulated by BRD9 KD plus pomalidomide compared to either alone. In addition, EGR1, a transcription factor downregulated to a greater extent by BRD9 KD plus pomalidomide compared to either alone, is necessary for p53-independent MYC-induced programmed cell death; EGR1 expression is positively correlated with growth and survival [43]. Further, EGR2 was more downregulated in response to BRD9 KD plus pomalidomide IMiD compared to either alone. EGR2, like EGR1, also plays a role in TNF-alpha signaling [46,47], another top downregulated pathway revealed by enrichment analysis to be associated with BRD9 KD plus pomalidomide. 

Major upregulated pathways revealed for BRD9 KD plus pomalidomide treatment include the immune system process, cell adhesion, and inflammatory response. GIMAP4, which is an enzyme of the GTP-binding superfamily and also a member of the immuno-associated nucleotide subfamily of nucleotide-binding proteins that play a role in the immune response [48], was upregulated more in response to doxycycline-induced BRD9 KD combined with pomalidomide. The combination stimulated the transcription of genes associated with STAT2 activity. The STAT2 signaling pathway, which plays a role in the immune response that includes cancer initiation and inflammation [49], was identified as a major upregulated pathway in response to doxycycline-induced BRD9 KD coupled with pomalidomide. 

Our MM cell line model of acquired resistance suggests that the addition of a BRD9-targeting drug to an IMiD can overcome IMiD resistance. CRBN protein levels were observed to be decreased in our drug-resistant cells, whereas MYC and IKZF3 protein levels were upregulated, similar to IMiD-resistant MM patient cells [11,50,68]. Consistent with these observations, we found that the combination of BRD9 targeting plus IMiD treatment was efficacious in reducing cell growth and led to a downregulation of MYC and IKZF3 and the upregulation of CRBN. Thus, it is likely that the overriding of the drug resistance phenotype characterized by altered MYC, IKZF3, and CRBN regulation may at least in part account for the effectiveness of BRD9 targeting plus pomalidomide in overriding drug resistance. As an approach to overriding IMiD resistance in MM, one study proposed combining pomalidomide and the HDAC6-selective inhibitor, A452, which similarly caused increased CRBN expression and decreased MYC and IKZF1/3 expression [69].

Our findings also support previous reports showing that CRBN is necessary for IMiD activity in MM, and its activity and expression are essential [7,51,70]. Upregulation of CRBN in MM cells would, therefore, be anticipated to sensitize MM cells to IMiDs and potentially re-sensitize IMiD-resistant MM cells to these drugs. Our previous work [26] suggests that BRD9 targeting leads to chemosensitization and leads to decreased MYC. Consistent with these findings, the combination of BRD9 targeting and IMiDs synergistically reduced the growth of MM cells and was associated with increased CRBN and decreased MYC. While the exact mechanism whereby CRBN is upregulated by a combination of BRD9 targeting and treatment with IMiDs is unclear, it is likely that chromatin remodeling, which influences transcriptional regulation, is involved at some level. 

For the studies presented herein, we were in the unique position to have a BRD9 inhibitor, EA-89, and its corresponding BRD9 degrader, QA-68 (EA-89, linked to a lenalidomide moiety), which allowed us to directly compare the ability of each agent to either inhibit or degrade BRD9 and deregulate target proteins or pathways of interest. QA-68 was observed to be more potent in terms of antileukemic activity than, e.g., the previously published BRD9 degrader, dBRD9a [56,57].

QA-68 and EA-89 were utilized as specific tools to define the impact of BRD9 targeting in combination with IMiDs on MM cell lines and primary cells in vitro. Ideally, we would like to expand our findings to efficacy studies and assess their toxicity in vivo as single agents and in combination with IMiDs. However, the chemical properties (solubility) and pharmacokinetics properties of these prototype molecules revealed challenges that limit their use for meaningful in vivo studies. The solubility of both QA-68 and EA-98 is less than 1 mg/mL in aqueous solutions, which is far below what is needed to support dosages for in vivo studies. In addition, we tested the in vitro clearance of QA-68 in mouse liver microsomes (MLM) and rat liver microsomes (RLM) and found that this compound exhibited high in vitro clearance (MLM-CL_int_ = 92 µL/min/mg and RLM-CL_int_ = 98 µL/min/mg). This suggests that high in vivo metabolism and clearance will further limit the use of these compounds for in vivo studies. 

We are currently working on strategies to overcome the limitations of these compounds in two parallel avenues. Specifically, we are developing derivatives of these compounds, which include chemical modifications, such as modified “linkers.” This strategy is likely to increase the solubility of the compounds but may also help to reduce in vivo metabolism and extend their half-life. Another strategy is to develop targeted nanoparticle-based formulations of these agents. In this case, both solubility and the pharmacokinetics (PK) of the formulation will depend mainly on the nanoparticle “carrier” rather than the properties of the encapsulated compounds. This strategy will not only overcome the solubility and PK limitations but will also improve the specificity of the delivery of the compounds to tumors, which has previously successfully been employed by others [71].

Taken together, our findings support the combination of a BRD9-targeting agent and IMiD as a promising new treatment approach for MM. Our results also suggest that the combination of BRD9 targeting and pomalidomide may be efficacious against IMiD-resistant MM.

## 5. Conclusions

The present study supports BRD9 as a therapeutic, druggable target for MM and introduces the novel concept of potentiation of the antitumor effects of IMiDs with BRD9 targeting. The observed synergy is associated with MYC and IKZF3 downregulation; Conversely, overexpression of these proteins can partly reverse synergy. MYC and E2F target genes, as well as pathways including cell cycle, cell division, and DNA replication, were revealed as being suppressed in association with synergy. In contrast, potentiation of IMiD treatment with BRD9 targeting was linked to the upregulation of pathways, including immune and inflammatory response and cell adhesion. Our findings support a robust dependency in MM on both BRD9 targeting and IKZF3 protein degradation, as simultaneous inhibition of both leads to marked MM cell killing with substantially higher potency than BRD9 targeting or IMiD treatment alone. Of significance, the combination of BRD9 targeting and IMiD treatment, which downregulates MYC and upregulates CRBN, was able to override IMiD resistance. These results suggest a novel and potentially highly efficacious therapeutic strategy for MM and IMiD-resistant MM.

## Figures and Tables

**Figure 1 cancers-16-01319-f001:**
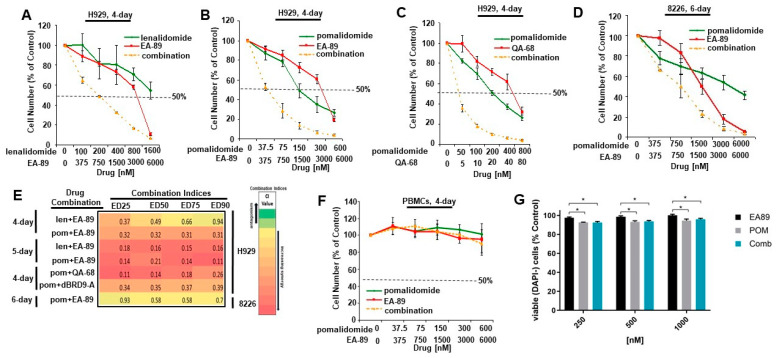
BRD9 inhibitor or degrader treatment potentiates effects of IMiDs on MM cell proliferation. (**A**,**B**) Proliferation assays: Effects of lenalidomide (**A**) or pomalidomide (**B**) alone or combined with the BRD9 inhibitor, EA-89, against H929 cells following 4 days. (**C**) Proliferation assay: Effects of pomalidomide alone or combined with the BRD9 degrader, QA-68, against H929 cells following 4 days. (**D**) Proliferation assay: Effects of pomalidomide alone or combined with EA-89 against 8226 cells following 6 days. Results shown here are representative of additional studies for which similar results were observed. (**E**) Combination indices generated by Calcusyn software for proliferation studies investigating combinations of BRD9-targeting agents and IMiDs. (**F**,**G**) Treatment of normal PBMCs following 4 days (**F**) or CD45+ leukocyte sub-population of PBMCs following 5 days (**G**)treatment with pomalidomide alone, EA-89 alone, or a combination of both agents (*n* = 6). Paired *t*-tests showing statistical significance *p* < 0.05 are denoted by * between the biological treatment groups EA89 alone and pomalidomide alone/combination but no significance observed between pomalidomide alone and combination.

**Figure 2 cancers-16-01319-f002:**
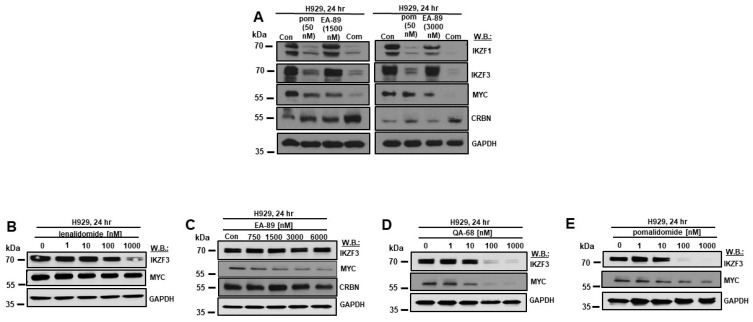
Effects of treatment of MM cells with IMiDs or BRD9 inhibitor or degrader, alone or in combination, on Ikaros, MYC, and CRBN protein levels. (**A**) Immunoblots: Effects of pomalidomide or EA-89 alone or combined on IKZF1, IKZF3, MYC, and CRBN protein levels following 24 h. (**B**–**E**) Immunoblots: Effect of lenalidomide (**B**), EA-89 (**C**), QA-68 (**D**), and pomalidomide (**E**) on levels of IKZF3 and MYC protein levels following 24 h. CRBN protein levels were analyzed in EA-89-treated cells (**C**). The uncropped blots are shown in Appendix A.

**Figure 3 cancers-16-01319-f003:**
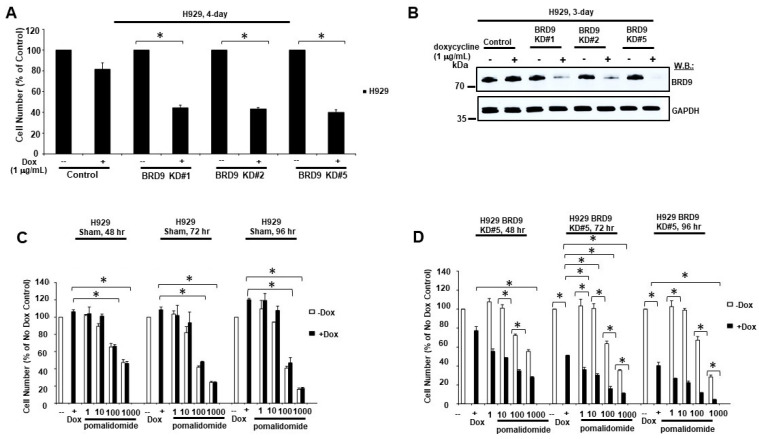
BRD9 KD potentiates effects of pomalidomide against growth of MM cells. (**A**) Proliferation assay: Effects of doxycycline-induced BRD9 KD on growth of H929 cells. For (**A**), asterisks indicate statistical significance (*p* value < 0.05). (**B**) Immunoblot: Measurement of BRD9 KD efficiency in H929 BRD9 KD#1, BRD9 KD#2, and BRD9 KD#5 cell lines versus H929 control cells. Doxycycline-inducible BRD9 KD H929 cells (KD#1, KD#2, and KD#5) were compared with H929 control cells in terms of growth response to treatment with doxycycline plus or minus pomalidomide across the indicated range of concentrations for 2 days, 3 days, and 4 days. For (**C**,**D**), asterisks indicate statistical significance (*p* value < 0.05). Results shown (**C**,**D**) are representative of other studies for which similar results were observed. The uncropped blots are shown in Appendix A.

**Figure 4 cancers-16-01319-f004:**
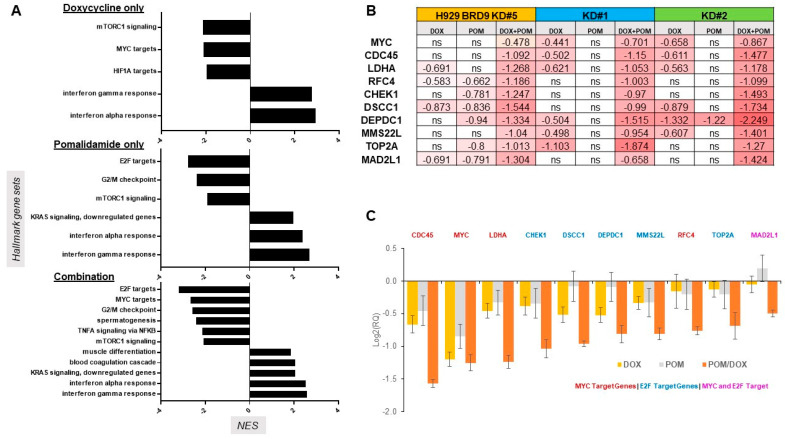
GSEA enrichment analysis and PCR validation of MYC and E2F target genes. (**A**) GSEA enrichment analysis of RNA-seq results for H929 BRD9 KD#5 cells treated with doxycycline, pomalidomide, or a combination following 24 h. (**B**) Heatmap showing representative MYC and E2F targets downregulated in RNA-seq from the three sets (namely, doxycycline, pomalidomide, or a combination) in H929 BRD9 KD#1, 2, and 5 cells. (**C**) PCR validation of selected MYC and E2F target genes associated with BRD9 KD plus pomalidomide treatment in H929 BRD9 KD#2 cells following 24 h.

**Figure 5 cancers-16-01319-f005:**
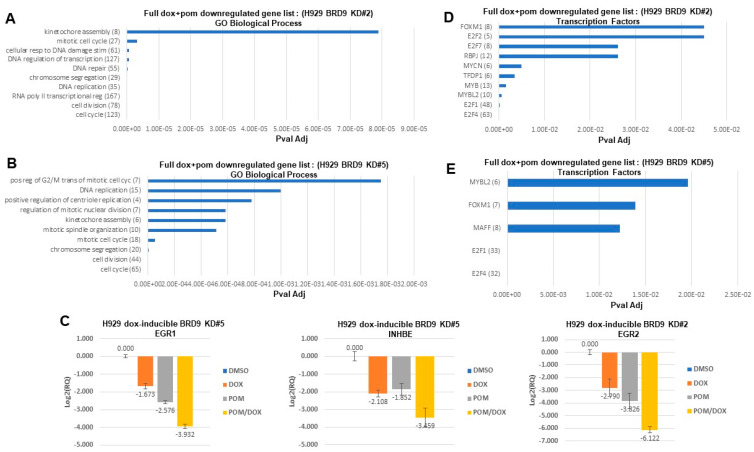
Inhibitory effects of pomalidomide combined with targeted loss of BRD9 on signaling pathways in MM. (**A**,**B**,**D**,**E**) Pathway analyses showing signaling pathways downregulated by doxycycline-induced BRD9 KD+pomalidomide treatment of H929 cells (clones #2 and #5) for 24 h. The full list of genes downregulated in response to doxycycline+pomalidomide was used for these analyses. Gene numbers are shown in parentheses. For (**A**,**B**), GeneCodis was utilized for pathway analysis, using GO Biological Process, and for (**D**,**E**), GeneCodis was utilized for pathway analysis, using Transcription Factors. (**C**) Validation (via qPCR) performed for EGR1, INHBE, and EGR2, downregulated in response to pomalidomide+doxycycline-induced BRD9 KD in H929 cells. Graphs shown are representative of replicates for which similar results were observed (KD#1 (EGR1, *n* = 2; INHBE, *n* = 1); KD#2 (EGR1, *n* = 2; EGR2, *n* = 2); KD#5 (EGR1, *n* = 2; INHBE, *n* = 1). Treatments were as follows: DMSO vehicle, doxycycline (1 μg/mL), pomalidomide (50 nM), and combination of doxycycline (1 μg/mL)+pomalidomide (50 nM). For doxycycline-treated H929 doxycycline-inducible BRD9 KD#1 cells, BRD9 KD#2 cells, and BRD9 KD#5 cells, BRD9 was downregulated Log2-fold −1.14, Log2-fold −0.53, and Log2-fold −1.02, respectively. MYC target genes include E2F transcription factors.

**Figure 6 cancers-16-01319-f006:**
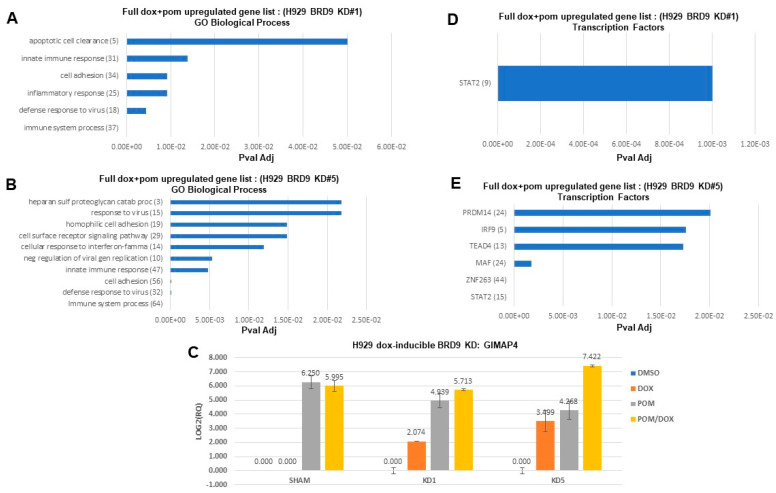
Stimulatory effects of pomalidomide combined with targeted loss of BRD9 on signaling pathways in MM. (**A**,**B**,**D**,**E**) Pathway analyses showing signaling pathways upregulated by doxycycline-induced BRD9 KD+pomalidomide treatment of H929 cells (clones #1 and #5) for 24 h. The full list of genes downregulated in response to doxycycline+pomalidomide was used for these analyses. Gene numbers are shown in parentheses. For (**A**,**B**), GeneCodis was utilized for pathway analysis, using GO Biological Process, and for (**D**,**E**), GeneCodis was utilized for pathway analysis, using Transcription Factors. (**C**) Validation (via qPCR) performed for GIMAP4, upregulated in response to pomalidomide+doxycycline-induced BRD9 KD in H929 cells. Treatments were as follows: DMSO vehicle (24 h), doxycycline (1 μg/mL), pomalidomide (50 nM), and combination of doxycycline (1 μg/mL)+pomalidomide (50 nM). For doxycycline-treated H929 doxycycline-inducible BRD9 KD#1 cells, BRD9 KD#2 cells, and BRD9 KD#5 cells, BRD9 was downregulated Log2-fold −1.14, Log2-fold −0.53, and Log2-fold −1.02, respectively.

**Figure 7 cancers-16-01319-f007:**
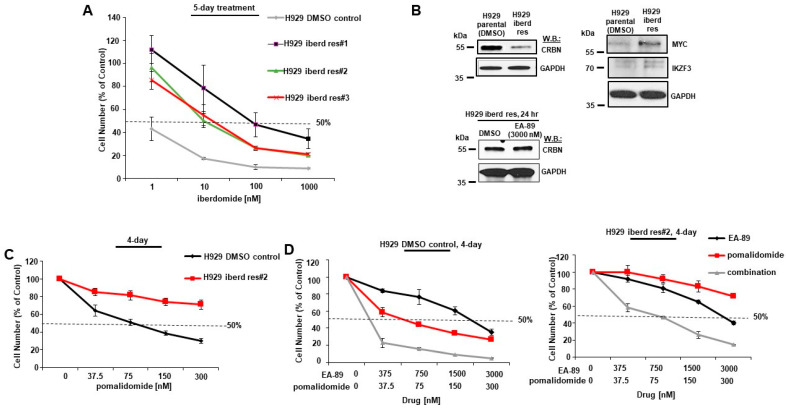
Overriding IMiD resistance with BRD9 targeting combined with IMiD treatment. (**A**) Proliferation assay: Effects of iberdomide against parental H929 cells (treated with DMSO for 4 months) or IMiD-resistant H929 cells (treated for 4 months with 10–20 nM iberdomide) following 5 days. (**B**) Immunoblots: (upper panel) Comparison of levels of CRBN, MYC, and IKZF3 protein in parental H929 cells (treated with DMSO for 4 months) or IMiD-resistant#2 H929 cells (treated for 4 months with 10 nM iberdomide); (lower panel) Effects of EA-89 (3000 nM) treatment of H929 iber-domide-resistant cells on CRBN levels following 24 h. (**C**) Proliferation assay: Effects of pomalidomide against parental H929 cells (treated with DMSO for 4 months) or IMiD-resistant H929 cells (treated for 4 months with 10–20 nM iberdomide) following 4 days. (**D**) Proliferation assays: (left panel) Effects of pomalidomide alone or combined with the EA-89 against H929 cells (treated with DMSO for 4 months) following 4 days. (right panel) Effects of pomalido-mide alone or combined with the EA-89 against H929 IMiD-resistant H929 cells (treated for 4 months with 10–20 nM iberdomide) following 4 days. The uncropped blots are shown in Appendix A.

## Data Availability

RNA-seq data generated during and analyzed during the current study will be submitted to Sequence Read Archive (SRA) and made available.

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
