# Peer review of "Synergy between BRD9- and IKZF3-Targeting as a Therapeutic Strategy for Multiple Myeloma"

_cancers, 2024, doi:10.3390/cancers16071319_

Round 1
Reviewer 1 Report (Previous Reviewer 1)
Comments and Suggestions for Authors
The authors addressed the comments and improved the manuscript.
Author Response
We thank the reviewer for the feedback and peer review.
Reviewer 2 Report (Previous Reviewer 2)
Comments and Suggestions for Authors
Thanks for author response. Author try to response the reviewer query but here alarming the compound solubility and challenges part is formulation. If authors mentioned hypothetical approaches not working , what are the backup strategies are available author need to mention. Author need to mentioned any advantage over already available degrader.
Author Response
We thank the reviewer for the feedback. Our responses are included in the attached document.

Round 2
Reviewer 2 Report (Previous Reviewer 2)
Comments and Suggestions for Authors
Now it will proceed for publication process.
This manuscript is a resubmission of an earlier submission. The following is a list of the peer review reports and author responses from that submission.
Round 1
Reviewer 1 Report
Comments and Suggestions for Authors
In this paper, the authors demonstrate that combination of IMiDs and BRD9 inhibitor/degrader exhibits anti-myeloma effect. The authors also show that this combination could overcome IMiDs-resistance in myeloma cells. These results are interesting, but there are a few points that need to be addressed.
Comments:
1. The authors used only one cell line (H929). This reviewer could not find the data of RPMI8226 cells even in the supplementary data. They should show the data of RPMI8226 cells in main figures and also should use other cell line to support that this combination exhibits synergistic effect in myeloma cells. This combination shows modest cytotoxic effect in primary myeloma cells.
2. What is the potential mechanism of CRBN upregulation by this combination? This should be discussed.
3. CRBN expression after EA-89 treatment should be shown in IMiDs-resistant cells (not parental cells).
Author Response
Point-By-Point Response to Reviewers Comments for Manuscript
We thank the reviewers for their constructive and thoughtful comments. We have carefully addressed these concerns and made appropriate changes wherever appropriate. Please find below a point-by-point response to each of the reviewers’ comments:
Comments by Reviewer 1:
- The authors used only one cell line (H929). This reviewer could not find the data of RPMI-8226 cells even in the supplementary data. They should show the data of RPMI-8226 cells in main figures and also should use other cell line to support that this combination exhibits synergistic effect in myeloma cells. This combination shows modest cytotoxic effect in primary myeloma cells.
Response: We agree and have now moved data performed in RPMI-8226 previously shown in supplemental data to the main manuscript. The combination study results for RPMI-8226 are now shown in Figure 2D. In Figure 2E, combination indices are shown for combination studies performed for both H929 and RPMI-8226 cell lines.
We added the following statements to describe these data:
Methods:
Cell lines and cell culture
H929 and RPMI-8226 MM cell lines were obtained from Dr. Kenneth Anderson (Dana-Farber Cancer Institute, Boston, MA).
Results:
3.1 BRD9 inhibitor or degrader treatment potentiates effects of IMiDs
… To test this notion, H929 or RPMI-8226 cells were treated with lenalidomide or pomalidomide alone, and combined with either the BRD9 inhibitor, EA-89, or the BRD9 degrader, QA-68. Consistent with previously reported findings [26], EA-89 or QA-68 treatment potentiated the growth inhibitory effects of lenalidomide or pomalidomide in both lines (Figure 2A-D and Supplementary Figure 1A-D), with Calcusyn combination indices suggestive of synergy across a range of drug concentrations (Figure 2E).
Discussion:
Using two established MM cell lines, H929 and RPMI-8226, we show that targeting of IKZF1/3 and BRD9, two established dependencies in MM, leads to synergistic effects on viability.
Figure Legends:
Figure 2. BRD9 inhibitor or degrader treatment potentiates effects of IMiDs. (A-B) Proliferation assays: Effects of lenalidomide alone, EA-89 alone, or the combination (A) or pomalidomide alone, EA-89 alone, or the combination (B) against H929 cells. (C) Proliferation assay: Effects of pomalidomide alone, QA-68 alone, or the combination against H929 cells. (D) Proliferation assay: Effects of pomalidomide alone, EA-89 alone, or the combination against RPMI-8226 cells. (E) Combination indices generated by Calcusyn software for proliferation studies investigating combinations of BRD9-targeting agents and IMiDs.
- What is the potential mechanism of CRBN upregulation by this combination? This should be discussed.
Response: Our previous work (Weisberg et al., 2022) suggests that BRD9 targeting leads to chemo-sensitization and leads to decreased MYC. Consistent with these findings, the combination of BRD9 targeting and IMiDs synergistically reduces the growth of MM cells and was associated with increased CRBN and decreased MYC. While the exact mechanism whereby CRBN is upregulated by a combination of BRD9 targeting and treatment with IMiDs is unclear, it is likely that chromatin remodeling, which influences transcriptional regulation, is involved at some level.
We added the following statements to the Discussion:
Our MM cell line model of acquired resistance suggests that addition of a BRD9-targeting drug to an IMiD can overcome IMiD resistance. CRBN protein levels were observed to be decreased in our drug-resistant cells, whereas MYC and IKZF3 protein levels were upregulated, similar to IMiD-resistant MM patient cells [11] [51] [67]. Consistent with these observations, we found that the combination of BRD9 targeting plus IMiD treatment was efficacious in reducing cell growth and led to a downregulation of MYC and IKZF3 and upregulation of CRBN. Thus, it is likely that the reversal of the drug resistance phenotype characterized by altered MYC, IKZF3 and CRBN regulation, may at least in part account for the effectiveness of BRD9 targeting plus pomalidomide in overriding drug resistance. As an approach to overriding IMiD resistance in MM, one study proposed combining pomalidomide and the HDAC6-selective inhibitor, A452, which similarly caused increased CRBN expression and decreased MYC and IKZF1/3 expression [68].
Our findings also support previous reports showing that CRBN is necessary for IMiD activity in MM and its activity and expression are essential [7] [69] [52]. Upregulation of CRBN in MM cells would therefore be anticipated to sensitize MM cells to IMiDs and potentially re-sensitize IMiD-resistant MM cells to these drugs. Our previous work [26] suggests that BRD9 targeting leads to chemosensitization and leads to decreased MYC. Consistent with these findings, the combination of BRD9 targeting and IMiDs synergistically reduces the growth of MM cells and was associated with increased CRBN and decreased MYC. While the exact mechanism whereby CRBN is upregulated by a combination of BRD9 targeting and treatment with IMiDs is unclear, it is likely that chromatin remodeling, which influences transcriptional regulation, is involved at some level.
- CRBN expression after EA-89 treatment should be shown in IMiDs-resistant cells (not parental cells).
Response: We carried out treatment of IMiD-resistant H929 cells with 3000 nM EA-89 for 24 hours and results suggest no measurable increase in CRBN protein levels. These results are now shown in a revised Figure 8B, lower panel. Our studies investigating EA-89 treatment of parental H929 cells generally also showed no increased CRBN protein expression when EA-89 was used as a single agent. We have added new CRBN expression results in a revised Figure 2 for panel 2I (EA-89 treatment), which clearly shows no change in CRBN expression following EA-89 treatment of parental H929 cells.
We have included the following statements to describe these data:
Results:
Description of results shown in Figure 2I: “Of note, EA-89 was not observed to influence CRBN protein levels (Figure 2I).”
Description of results shown in Figure 8B: “We did not observe EA-89 as a single agent to affect levels of CRBN protein in parental or iberdomide-resistant H929 cells (Figure 2I and Figure 8B, lower panel).”
Figure Legends:
Figure 2. BRD9 inhibitor or degrader treatment potentiates effects of IMiDs. (H-K) Immunoblots: Protein levels were measured as indicated in cells treated with lenalidomide (H), EA-89 (I), QA-68 (J) and pomalidomide (K).
Figure 8. Overriding IMiD resistance with BRD9 targeting combined with IMiD treatment. (B) Immunoblots: (Upper panel) Comparison of levels of CRBN, MYC, and IKZF3 protein in parental H929 cells (treated with DMSO for four months) or iberdomide-resistant#2 H929 cells (treated for four months with 10 nM iberdomide). (Lower panel) Effects of EA-89 (3000 nM) treatment of iberdomide-resistant H929 cells on CRBN levels.
Comments by Reviewer 2:
- In the present manuscript author try to describe a novel concept and treatment strategies of multiple myeloma. Furthermore, they are target with two novel protein targets for beneficial treatment such as BRD9 and IKZF3. Overall manuscript flows look good and impressive compilation but few concerns need to address before further processing such as author need to clarify patient selection criteria for beneficial this strategy with proper justification.
Response: Both normal and myeloma bone marrow aspirate samples (relapsed/refractory) were used. The primary specimens were purchased from BioIVT (Westbury, New York):
https://bioivt.com/human-bone-marrow-aspirate
https://bioivt.com/human-heme-oncology-bone-marrow
We have now included the following statement in the Methods:
“Both normal and myeloma bone marrow aspirate samples (relapsed/refractory) were used. The primary specimens were purchased from BioIVT (Westbury, New York).”
- Also several preclinical studies (Syngenic, Xenograft , PDX ) are missing here and this type of study will helpful for correlation the current outcome. Any toxicity profiling and pharmacokinetic studies are appreciable to improve the manuscript quality.
Response: We appreciate the reviewer’s suggestion. QA-68 (BRD9 degrader) and EA-89 (BRD9 inhibitor) were utilized primarily as a chemical probe to demonstrate the impact of BRD9 targeting in combination with IMiDs on MM cell lines and primary cells in vitro. Ideally, we would want to demonstrate their efficacy and assess their toxicity in vivo as single agents and in combination with IMiDs.
However the chemical properties (solubility) and pharmacokinetics properties of these prototype molecules revealed multiple challenges that limits their use for meaningful in vivo studies and modifications are beyond the scope of this manuscript:
Solubility: The solubility of both QA-68 and EA-98 is less than 1 mg/ml in aqueous solutions, which is far below that need to support dosages for in vivo studies.
Pharmacokinetics in vitro: We tested the in vitro clearance of QA-68 in mouse liver microsomes (MLM) and rat liver microsomes (RLM). Unfortunately, we found that this compound exhibited high in vitro clearance (MLM-CLint= 92 µL/min/mg and, RLM-CLint =98 µL/min/mg). Which suggests high in vivo metabolism and clearance that limits even further the use of these compounds for in vivo studies.
We are currently working on strategies to overcome the solubility and the pharmacokinetic limitations of these compounds or derivatives in two parallel avenues:
- We are developing the next generation of these compounds, which include chemical modifications and conjugation with different soluble “linkers”, such as polyethylene glycol. This strategy will not only increase the solubility of the compounds, but also may help them avoid liver metabolism and extend their T1/2 in the blood.
- We are developing targeted nanoparticle-based formulations of these compounds alone or co-encapsulated with IMiDs. In this case, both solubility and the pharmacokinetics of the “formulation” will depend mainly on the nanoparticle “carrier”, rather than the properties of the encapsulated compounds. This startegy will not only overcome the solubility and PK limitations, but will also improve the specificity of the delivery of the compounds to tumors.

Reviewer 2 Report
Comments and Suggestions for Authors
In the present manuscript author try to describe a novel concept and treatment strategies of multiple myeloma. Furthermore, they are target with two novel protein targets for beneficial treatment such as BRD9 and IKZF3. Overall manuscript flows look good and impressive compilation but few concerns need to address before further processing such as author need to clarify patient selection criteria for beneficial this strategy with proper justification. Also several periclinal studies ( Syngenic, Xenograft , PDX ) are missing here and this type of study will helpful for correlation the current outcome. Any toxicity profiling and pharmacokinetic studies are appreciable to improve the manuscript quality.
Author Response

(The authors gave the same response as above.)

Round 2
Reviewer 1 Report
Comments and Suggestions for Authors
The author improved the manuscript.
Reviewer 2 Report
Comments and Suggestions for Authors
Accepted in present form.